# A Holistic Approach for the Identification of Success Factors in Secondary Cleft Osteoplasty

**DOI:** 10.3390/jpm12030506

**Published:** 2022-03-21

**Authors:** Tom A. Schröder, Martin Maiwald, Axel Reinicke, Uwe Teicher, André Seidel, Thorsten Schmidt, Steffen Ihlenfeldt, Karol Kozak, Winnie Pradel, Günter Lauer, Anas Ben Achour

**Affiliations:** 1Department of Oral and Maxillofacial Surgery, University Hospital Carl Gustav Carus Dresden, Technische Universität Dresden, Fetscherstraße 74, 01307 Dresden, Germany; winnie.pradel@uniklinikum-dresden.de (W.P.); guenter.lauer@uniklinikum-dresden.de (G.L.); 2Else Kröner Fresenius Center for Digital Health, Technische Universität Dresden, Fetscherstraße 74, 01307 Dresden, Germany; 3Chair of Material Handling, Technische Universität Dresden, Münchner Platz 3, 01187 Dresden, Germany; martin.maiwald@tu-dresden.de (M.M.); thorsten.schmidt@tu-dresden.de (T.S.); 4Institut für Angewandte Informatik e.V., Schnorrstraße 70, 01069 Dresden, Germany; reinicke@infai.org (A.R.); kozak@infai.org (K.K.); 5Fraunhofer Institute for Machine Tools and Forming Technology IWU, Nöthnitzer Straße 44, 01187 Dresden, Germany; uwe.teicher@iwu.fraunhofer.de (U.T.); andre.seidel@iwu.fraunhofer.de (A.S.); or steffen.ihlenfeldt@tu-dresden.de (S.I.); 6Chair of Machine Tools Development and Adaptive Controls, Technische Universität Dresden, Helmholtzstraße 7a, 01069 Dresden, Germany

**Keywords:** osteoplasty, cleft palate, bone grafting, experimental data, data integration, holistic data chain, medical data management, clinical databases

## Abstract

Cleft lip and palate belong to the most frequent craniofacial anomalies. Secondary osteoplasty is usually performed between 7 and 11 years with the closure of the osseus defect by autologous bone. Due to widespread occurrence of the defect in conjunction with its social significance due to possible esthetic impairments, the outcome of treatment is of substantial interest. The success of the treatment is determined by the precise rebuilding of the dental arch using autologous bone from the iliac crest. A detailed analysis of retrospective data disclosed a lack of essential and structured information to identify success factors for fast regeneration and specify the treatment. Moreover, according to the current status, no comparable process monitoring is possible during osteoplasty due to the lack of sensory systems. Therefore, a holistic approach was developed to determine the parameters for a successful treatment via the incorporation of patient data, the treatment sequences and sensor data gained by an attachable sensor module into a developed Dental Tech Space (DTS). This approach enables heterogeneous data sets to be linked inside of DTS, archiving and analysis, and is also for future considerations of respective patient-specific treatment plans.

## 1. Introduction

Cleft lip and palate are the most frequent craniofacial anomaly, with an incidence of 1 in 700 births [1]. The cleft is a result of incomplete fusion of soft and hard tissue during embryonic development. To achieve dental rehabilitation, restoration of form and function, the bony continuity of the dental arch must be rebuilt [2,3]. The keystone in cleft management is surgical treatment, and it varies depending on the cleft type. Closure of soft and hard tissue clefts is done in several surgical steps in different versions [4]. It is widely used to do the closure of the osseus defect before eruption of the permanent maxillary canine at the age of 7–11 [5]. Therefore, the autologous bone graft from the iliac crest is mainly used due to his osteoconductive, osteoinductive and osteogenic potential [6]. Shepard chisels and a trephine hand drill are used to harvest bone cylinders. The second step is the preparation of the bone cylinders, which is mainly done by hand. The processed cylinders are then inserted into the cleft defect by hand too.

With the help of the grafted bone, the cleft volume can be reduced by approximately 62% [7]. In short-term follow ups, about 4.4% of the patients show one or more complications [8]. Complications include mainly infections (1.1–6.6%), wound dehiscence (2.6–3.2%), fistula (0.7%), stitch granuloma (0.2%) and pressure necrosis (0.05%) [8,9,10]. Occasionally there is rejection of small cancellous bone particles 4–8 weeks post-op. The main goal of this concept is to gain a better understanding of the treatment process itself, and to improve it and reduce complications via recommendations for appropriate treatment, which were derived by a holistic data analysis.

The basis for this is a data analysis, data management of the surgical techniques and treatments with reference to historical data. Thus, a field of relevant success parameters of the treatments will be identified and established. These parameters are to be acquired for future treatments using data capture in the database, objective modern sensor technologies and related to medical treatment parameters. The necessary data management platform, DTS for the existing heterogeneously structured data sets represents the interface for data acquisition, management, archiving and analysis.

## 2. Materials and Methods

### 2.1. Analysis of Retrospective Data

This study was approved by the local ethics committee. The retrospective data were acquired from the hospital information system ORBIS^®^ (DH Healthcare GmbH, Version 03.15.02.00, 53227 Bonn, Germany) and the dental database DENTWARE^®^ (DENTWARE Computer GmbH, Version 1.3n, 82216 Maisach, Germany). The data were analyzed with SPPS^®^ (IBM SPSS Statistics for Windows, Version 28.0. IBM Corp, Armonk, NY, USA). In collecting the medical records, the focus was on the surgical treatment of cleft lip and palate. The surgical techniques used to treat cleft lip and palate are varied and have a direct impact on the functional outcome. This leads to a core requirement for the medical database: The tool must capture generic parameters that allow different treatment techniques to be compared with each other.

To develop a suitable data structure (TRL 8), the treatment sequence at University Hospital Carl Gustav Carus in Dresden was recorded and combined into treatment packages. These are based on the LAHSAL classification for cleft lip and palate [11] and include:General procedures, which are performed regardless of the cleft type;Procedures that are performed to treat the cleft lip;Procedures that are specific to the cleft palate;Procedures that are specific to the alveolar cleft.

The scheduling of the procedures is based on the date of birth of the patient and takes into account earliest and latest times to perform them; time variance decreases as the treatment progresses with steps already performed. In addition, required personnel resources with execution durations are assigned to the operations, which can serve as a basis for subsequent personnel resource planning.

A key aspect for the subsequent analysis of the correlations between healing processes and treatment characteristics is the most detailed data possible. This includes preoperative general data (age, weight, height, gender, family anamnesis, type of cleft) and diagnostic data (X-ray images, Cone Beam Computed Tomography (CBCT)), operation-specific data (duration of surgery, duration of anesthesia and the tools used) and postoperative data describing the healing process (wound infection and dehiscence). This allows a much more detailed analysis for successive improvement of surgical techniques over a long term, which is not possible with the current state of the art [12].

### 2.2. Analysis and Definition of Process Parameters for Quantitative Measurement

To measure the mechanical loads during the removal of bone cylinders and the geometric adaptation to the cleft, it is necessary to analyze the tools and their principles of action in order to interlink to the enormous mechanical loads during processing [13] with the subsequent deterioration of the vitality of the bone. For the removal of the bone graft from the iliac crest, the tools shown in Figure 1 and Figure 2 are mainly used at the University Hospital in Dresden.

These tools differ significantly in their process characteristics. A comparison of process-relevant differentiation criteria is shown in Table 1. While the Shepard chisel stamps out the bone tissue with the help of hammer strokes, a cutting process occurs in the trephine hand drill due to the manually controlled rotation and feed movement of the mill. Once the target depth has been reached, the cylinder can be cut off via a rotary movement thanks to an integrated cutting groove. The bone cylinder is then removed from the chisel by using an ejector. However, using the trephine hand drill the bone cylinder must be detached from the remaining bone with additional instruments.

To describe the mechanical loads during the harvesting of bone augments, it is important to record the effective feed forces and the effective torque directly via the tool. With regard to clinical practice, the use of external sensors at the donor site is impossible due to practicability during operation and sterilization issues, so that an integration into the tool is necessary. Furthermore, it is currently not possible to describe the motion control. An additional recording of the accelerations and on the other hand the angular velocities in three-dimensional space enables the quantitative description of the tool movements during the harvesting process. Nevertheless, these sensors have to be integrated into the tool as well. During material separation, part of the energy is always converted into heat, which is why monitoring of the resulting temperature elevations near to the tool tip is considered necessary.

### 2.3. Method to Integrate Data from Different Sources

Data integration and data management are crucial issues in the environment of heterogeneous dental patient-process data sources. Considering various types of data, integration problems have to be solved in order to gain useful information and knowledge, while appropriate analytical methods have to be applied for comprehensive and uniform data. By merging interdisciplinary expert knowledge, the project team is developing a routine with data management methods for a standardization and optimization of surgical therapy for complex shaped bone defects using the example of cleft alveolus in a lab scale. Starting with 3D models generated from medical images, subsequently special physical training models are produced. Mechanical loads during bone graft removal and fitting have to be recorded by sensors locally.

The Dental Tech Space (DTS) database, proposed in this method, allows surgeons and production engineers a systematic data collection as a routine using web user interfaces. Using DTS, the surgeon collects the planned procedure on CBCT Images, patient data (linked with Hospital Information System (HIS)). DTS requires, at a minimum, an IIS (Internet Information Server) Web server, a SQL (Structured Query Language) database server and secure data storage on a server installed inside of hospital IT infrastructure. The data of Dental Tech Space are kept within on PostgreSQL, the relational tables of which follow the Entity–Attribute–Value model (EAV). This model is common in the medical record field and may have efficiencies with sparse data. In short, do not expect to find a tabular format (rows/columns or observations/variables). The database is initiated and managed by a group consisting of three academic staff:A physician familiar with clinical data management (Faculty of Health Sciences);A research scientist assigned as the DTS administrator (Faculty of Health Sciences);A computer scientist with experience in production, for software and server support (Information Engineering).

The DTS software keeps a record of concurrent users, daily user logins, projects created, projects moved to production, active users, first-time visitors, logged events and page hits. A practical and desirable way of running DTS is to set up these as virtual machines. Virtualization is a technology that allows one physical computer to run several “virtual” computers, which share the physical hosts’ resources. These virtual machines behave and appear both externally and internally as if they were real computers.

For reasons of generalizability and extensibility, patient data, production data and the implant database should be integratable [14], so that the surgeon can determine which bone implant respective surgical methods are most suited for the treatment of different disorders. Subsequently, the database generates a dashboard and a report that helps the operating staff for the preparation of the surgery with the selected implants based on production features. Moreover, the surgeon can use a report to prove his preparations for the procedure and to comprehensively explain the operation to the patient.

## 3. Results

### 3.1. Analysis Medical Data

A total of 49 patients were observed in the period between March 2005 to December 2021 at the University Hospital Dresden. The mean age at surgery was 11.7 (SD ± 4.2) years and 32 patients (65.3%) were male. There were 49.0% of the clefts localized on the left side, 24.5% right, 2.0% midline and 24.5% appeared on both sides. For cleft defects, 88.9% were filled with autologous iliac crest bone and 11.1% with tissue-engineered bone (autogenous osteoblasts cultured on demineralized bone matrix (Osteovit^®^, B. Braun, 34212 Melsungen, Germany)) [15]. The iliac crest bone was harvested to 71.8% from the left side. The mean time to harvest the iliac crest was 31.4 (SD ± 28.5) min, and the full closure of the cleft was 128.9 (SD ± 86.8) min. A total of 51.0% of the patients received a preoperative CBCT and 49.0% received one postoperative.

The database was expanded, especially through operation-specific data, which was not recorded before. Therefore, appropriately equipped tools are necessary and will be presented in the following section. The data structure was derived from the hierarchical master treatment plan adopting production and process planning methods. It contains 116 process elements, with a specific data set assigned to each step. Figure 3 shows the schematic generation of a patient-specific treatment plan with lifetime dates depending on the diagnosis and date of birth. The plan contains up to five layers with milestones in the top level—since the surgery of the cleft palate is the most complex step it covers all levels. The grey highlighted elements represent an example for the classification of the bone harvest in the treatment plan.

In this example, the database related to the step of the removal of the bone cylinders will be enhanced by tool data such as angles, rotational speeds, forces and torques, and this differs from previous approaches. In addition, the technical data durations and resources will be part of the dataset, which can be used for planning purposes in a bigger context (human resources, operating room planning, etc.). In the future the database has to take further treatment options into account to make it customizable.

### 3.2. Sensory Concept for In Vitro Experiments

Taking into account that there are different treatment approaches, a reliable and objective measurement system for relevant steps of the long-term treatment sequences needs to be established. The aim of the sensory recording is to identify success or complication factors for the harvesting process in order to assure high vitality of the bone transplants. This represents one critical process in the treatment sequence and the concept of plug-and-play sensor modules can be further extended to other critical processes, which are done manually at the current state. For the proof-of-concept, an external sensor module was developed based on commercially available hardware. The sensor module can be attached to the standard tools via integrated joining points. In order to enable a largely impairment-free application compared to the usual handling, a wireless measuring chain (sensor, measurement electronics, battery pack) for the mechanical loads was integrated into the sensor module. The data of those sensors are transferred to a PC via Bluetooth^®^ (2direct GmbH, Logilink Version 4.0, 58579 Schalksmühle, Germany). To measure the three-dimensional position profile, a standard chip with integrated accelerometer and gyroscope was fixed in a pocket of the sensor module in a defined orientation. This chip is connected to a Raspberry Pi via cable. An integration of standard temperature sensors in the tool wall near the tip is because of the small dimensions not possible. Thermopolymers, which irreversibly discolor when exceeding defined limit temperatures, are applied to the circumferential surface to record the maximum temperatures.

To validate the external sensor module, a preliminary experiment for the Shepard chisel was done. For that reason, a piezoelectric dynamometer with a very high sampling rate was used as a reference measuring system. As seen in Figure 4 the hammer stroke pulses are detected in a comparable pattern. Comparing the negative springback forces of the reference measurement system, it can be seen that the absolute values of these tensile forces are substantially lower than the thrust forces during the hammer strokes.

Differences in the retraction period between the reference measurement system and the sensor module can be addressed regarding the missing preload of the sensors of the external sensor system, the measurement principle with strain gauges and the technically limited sampling rate of this system. Nevertheless, this sensor prototype is capable of measuring the strokes during hammering exactly with the ability to handle the system manually in all spatial directions during operation.

The external sensor module proved to be an applicable tool for laboratory experiments on the correlation between harvesting process and vitality of bone cylinders and serves as a basis for future monitoring systems during bone grafting.

### 3.3. Specification of Dental Tech Space Software

Dental Tech Space presents an architecture that implements data integration in hospital from the production, surgery preparation and patient data. DTS integrates sub databases without any changes to the individual databases (SQL database, software backend, API, Frontend) or any need to treat another database. The solution combines database technology and a wrapper layer known from Extraction Transformation Loading (ETL) systems and bring it to SQL Database, WEB API (backend) layer, Interface layer (Rest API) and frontend. It also provides semantic integration through connection mechanism between data elements. The solution allows for integration of patient, surgery and production data in one technological software framework: data management platform and implementation of analytical methods in one end-user environment. The patient data (see Figure 5) are transferred, secured, to a Hospital Information System (HIS). Medical data storage in DTS offers a highly scalable clinic web storage service that uses cumulative digital objects (e.g., patient, surgery, implant) rather than blocks or files. An object is simply a piece of data in no specific format: it could be a file, an image, a piece of seismic data or some other kind of unstructured content. Object storage typically stores data, along with metadata that identifies and describes the content. For metadata management and automated quality control and data fusion (ETL Processes), a data consistency model (DTS metamodel) is used to enable eventual consistency for updates or deletions to existing objects. This means that if an existing object is overwritten, there is a chance that a re-read of that object may return a previous version, as replication of the object has not completed between machine laboratories in availability zones in the same region.

DTS provides an API (Application Programming Interface). The API Token generated restricts access to only the project for which the administrator created it, requiring the user to have access to that project.

DTS is a secure, web-based application that an SAS (Self Assessment Sheet) programmer should be able to use intuitively to build forms and export their data. In the most basic use, DTS provides a simple method to export the data from the forms of a project and to run a program that DTS generated to create temporary datasets and formats. Finally, to further reduce the manual interaction with the GUI (Graphical User Interface) an automatic guidance functionality is provided.

## 4. Discussion

The available literature ranges from expert opinion to multi-year field studies, but not all treatment techniques receive equal consideration, which makes them hard to compare due to the lack of data [12,16]. The current standard in cleft osteoplasty is largely dominated by the manual execution of the surgical steps. As a result, the success of the operation is very much dependent upon the skill of the surgeon [17]. To gain the full understanding of all treatment steps and avoid errors, a collection of all data is necessary. We noticed a lack of data performing the retrospective analysis. Postoperative complications are not documented in such a way that they can be reasonably analyzed to allow conclusions about bone healing. Different documentation systems are used. Documentation by hand and two digital documentation systems do not allow a direct overview.

After surgery, bone resorption in the augmented site occurs to a certain degree. The influence of the harvesting site has been described, but the influence of the harvesting process using a Shepard chisel or a trephine hand drill remains still unclear. Klijn et al. showed in a meta-analysis that the augmentation of the maxilla sinus floor had significantly less bone if the iliac crest was harvested compared to mandibular symphysis and ramus bone [18]. Similar results were shown regarding secondary cleft surgery in a meta-analysis by Kamal et al. [7]. The bone loss due to resorption one year after secondary cleft osteoplasty using iliac crest ranges between 43.1% and 49.5% [19,20]. The bone resorption is even higher if the canine tooth eruption is delayed or if it is missing and no gap closure is performed. However, there is no clear indication if the bone harvesting in terms of harvesting injury has any impact on bone resorption in cleft surgery. Additionally, the influence of the mechanical and thermal loads during the harvesting process is widely unknown. For that reason, sensory acquisition of single manual processes is a key to enable a holistic approach for multiscale treatment processes. A comprehensive retrofitting of standard tools with external sensor tools enables the discovery of previously unrecognized interrelationships of consecutive processes and enables evidence-based individual patient treatment concepts. To enable this retrofit, it is necessary to miniaturize the sensor tools and ensure complete wireless data transmission. The data generated in this way must be merged with other data sources such as patient or physiological data in specially developed software platforms. With the help of intelligent data evaluation algorithms, archived treatment sequences could be examined to identify complication cluster. The holistic documentation of treatment procedures and individual processes enables the root causes of complications to be narrowed down or to be identified more quickly.

## 5. Conclusions

The identification of success factors of manual processes and therapies with a large number of treatment steps is not possible according to the current state of the art. There exists a strong need to digitize the therapies in a holistic manner to identify correlations and generate protocols of best practice. The approach to digitize process sequences, process parameters of relevant steps of the surgery, via add-on sensor systems and the consolidation of medical, patient-specific and process data in a superordinate platform enables clinicians to do that. The approach presented in this work allows for the integration of other complex treatment sequences and extension to other surgical disciplines and can make a major contribution to minimizing complication rates based on a stable data basis.

## Figures and Tables

**Figure 1 jpm-12-00506-f001:**
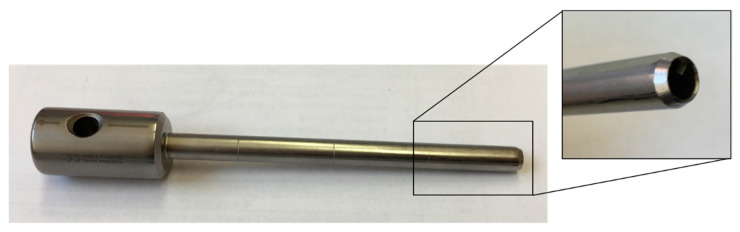
Shepard cylinder osteotome (chisel) used for harvesting of trabecular bone cylinders.

**Figure 2 jpm-12-00506-f002:**
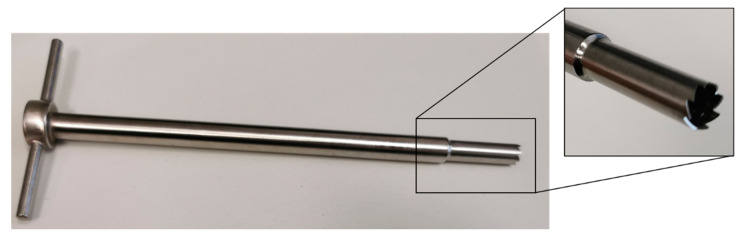
Trephine hand drill used for harvesting of trabecular bone cylinders.

**Figure 3 jpm-12-00506-f003:**
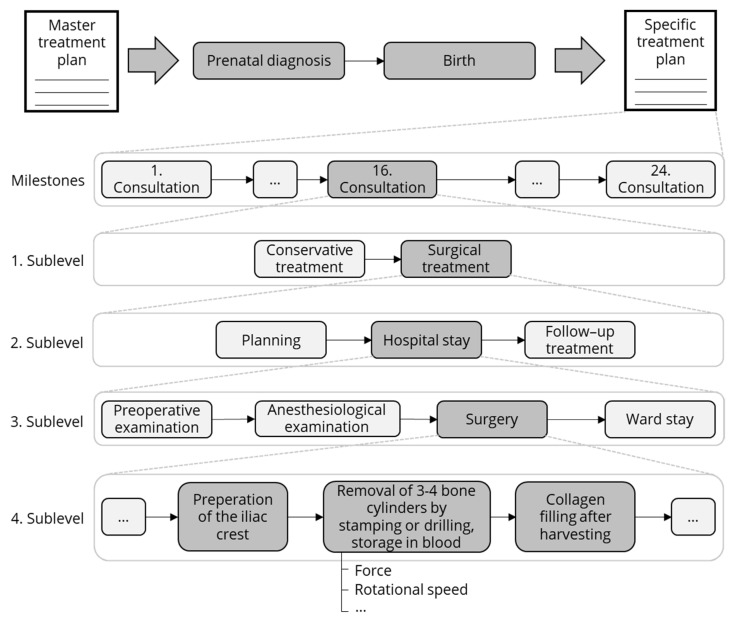
Process chart of bone harvesting.

**Figure 4 jpm-12-00506-f004:**
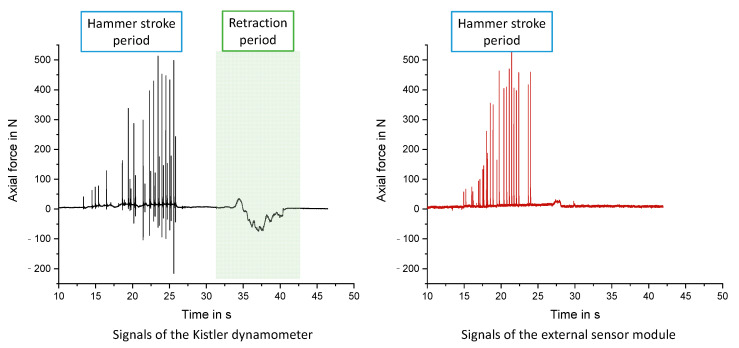
Comparison of the axial force measured with a high precision piezoelectric Kistler dynamometer and the self-developed external sensor module using a Shepard chisel.

**Figure 5 jpm-12-00506-f005:**
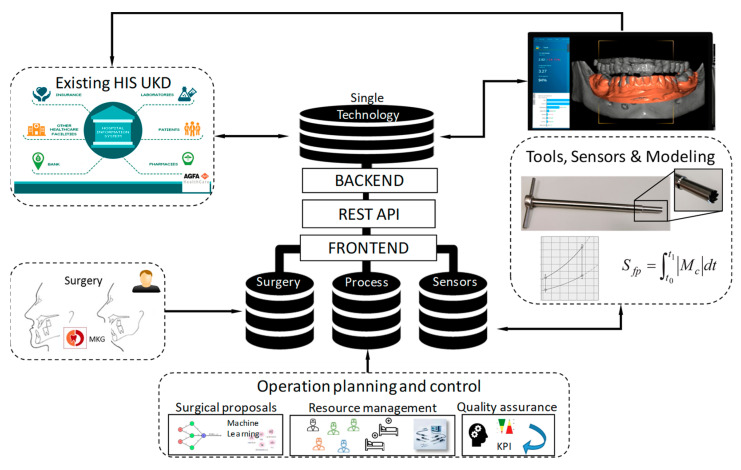
Architecture and data flow in Dental Tech Space (DTS).

**Table 1 jpm-12-00506-t001:** Differences in process characteristics of the used tools.

Criteria	Shepard Chisel (Figure 1)	Trephine Hand Drill (Figure 2)
Material separation process	stamping	machining–core drill
Feed movement	linear	helical (linear feed + rotary cutting movement)
Type of feed	impact (manual)	quasi-continuous (manual)
Auxiliary tool	hammer	-
Hole type	blind hole	blind or through hole

## Data Availability

The data presented in this study are available on request from the corresponding author. The data are not publicly available due to ethical restrictions.

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
