# Peer review of "A Holistic Approach for the Identification of Success Factors in Secondary Cleft Osteoplasty"

_jpm, 2022, doi:10.3390/jpm12030506_

Round 1

Reviewer 1 Report

Dear authors, 

this is a very interesting "Communication". 

Only small typos must be corrected in the text such as at lines 24, 260 in the squares, and please change very old references with newer ones, for examples number 5 and 11. 

Thank you 

Reviewer 2 Report

Thank you for your submission looking at data collection and analysis in cleft patients. It is well recognised that the management of cleft lip and palate is a long-term challenge. In order to be able to better assess our results and hence improve outcomes, there needs to be a holistic approach to data collection in a way that is reliable, repeatable and accessible in the future.

Multiple databases have been developed historically, the one described in tis paper has promise as a comprehensive way to collect and analyse the data.

Despite looking at a holistic database, the paper then focusses on analysis of a particular surgical procedure, namely the harvesting of iliac crest bone graft. This highlights on e of the challenges in measurement. The paper describes two different techniques for harvesting bone. There are also other described techniques used, making direct comparison of outcomes challenging. The more complete the data, however, the better the results can be assessed.

The database described is of interest, as this has been a long term challenge in may centres. The specific measurement of forces when harvesting the bone graft not so much. The technique that you describe is not universal.

I would recommend rewriting this paper with more emphasis on the database as that is more likely to be of interest to the readers than the pressure measurements when harvesting bone graft.

It will be interesting to see in the future, what findings there are from teh analysis of data collected using the sensors attached to the bone harvesting instruments and whether there are any correlations.

Of more interest would possibly be the findings when the entirety of the data is analysed. Larger numbers of cases would, however, likely be required for results to reach statistical significance which would necessitate multi-institution cooperation and data entry.

Round 2

Reviewer 2 Report

Thank you for your responses. the changes you have made are good.